# Frequency Autoregressive Image Generation with Continuous Tokens

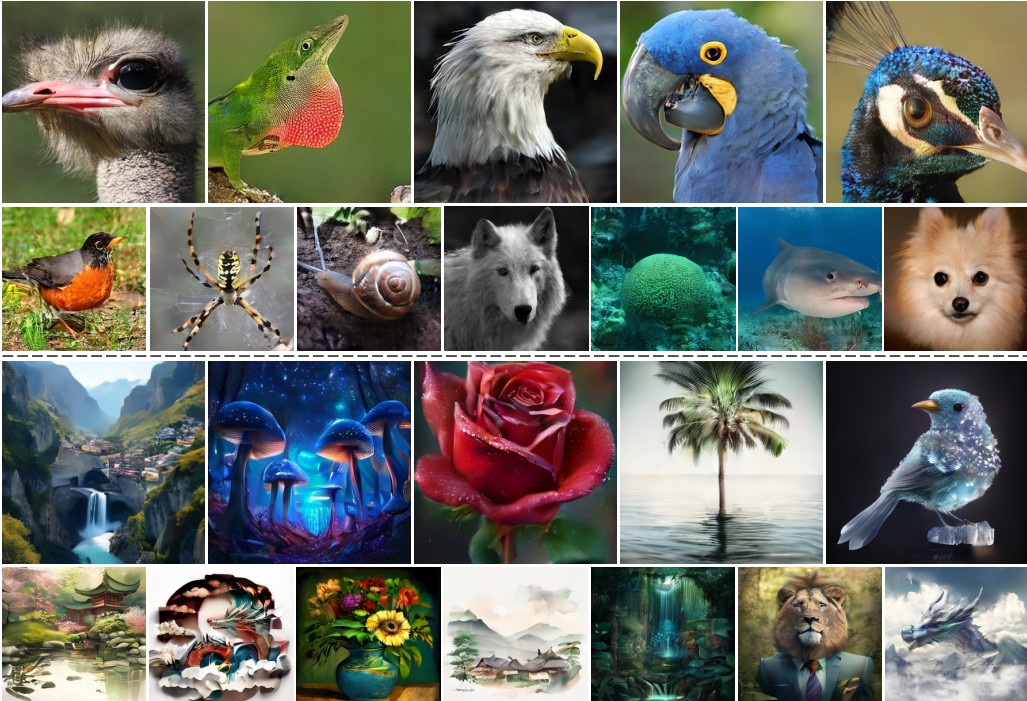

Figure 1: **Samples from our FAR autoregressive model with continuous tokens.** Upper part: class-conditional generation. Lower part: text-to-image generation (with prompts in Section J of the Appendix). All of these samples are generated with only 10 steps.

## Abstract

Autoregressive (AR) models for image generation typically adopt a two-stage paradigm of vector quantization and raster-scan "next-token prediction", inspired by its great success in language modeling. However, due to the huge modality gap, image autoregressive models may require a systematic reevaluation from two perspectives: tokenizer format and regression direction. In this paper, we introduce the frequency progressive autoregressive (**FAR**) paradigm and instantiate FAR with the continuous tokenizer. Specifically, we identify spectral dependency as the desirable regression direction for FAR, wherein higher-frequency components build upon the lower one to progressively construct a complete image. This design seamlessly fits the causality requirement for autoregressive models and preserves the unique spatial locality of image data. Besides, we delve into the integration of FAR and the continuous tokenizer, introducing a series of techniques to address optimization challenges and improve the efficiency of training and inference processes. We demonstrate the efficacy of FAR through comprehensive experiments on the ImageNet dataset and verify its potential on text-to-image generation.

# 1 INTRODUCTION

Building upon autoregressive models, large language models (LLMs) (Devlin, 2018; Raffel et al., 2020; Brown, 2020; OpenAI, 2022) have unified and dominated language tasks with promising intelligence in generality and versatility. Resembling language processing, a typical AR paradigm for image generation involves two stages: **1)** Discretizing image data via vector quantization (VQ) with a finite, discrete vocabulary; and **2)** Flattening the quantized tokens into a 1-D sequence for next-token prediction. Based on this foundational paradigm, recent works Chang et al. (2022); Yu et al. (2023b; 2024b); Tian et al. (2024); Han et al. (2024); Sun et al. (2024); Li et al. (2024) have made inspiring advancements in image generation.

However, due to the huge modality gap between vision and language data, directly inheriting autoregressive generation from language to image is far from optimal. Text and image represent two distinct modalities: **1)** Text is discrete, causal/sequential, and arranged in 1-D; **2)** Image is continuous, non-causal/non-sequential, and arranged in 2-D. These differences introduce two crucial considerations for autoregressive image generation **1)** *Tokenizer format* (concrete vs. continuous); **2)** *Regression direction* (incorporating image-specific causality). Regarding the tokenizer format, continuous tokenizer aligns more naturally with image data and induces less information loss Li et al. (2024); Fan et al. (2024); Rombach et al. (2022) (more analyses are placed in Section B of the Appendix). For the regression direction, raster Sun et al. (2024) or random Li et al. (2024) order fails to establish a causal sequence for images and can undermine inherent data priors, such as spatial locality. Consequently, the current AR paradigm for image generation remains sub-optimal and necessitates further investigation.

In this paper, we rethink the autoregressive image generation paradigm from a *spectral dependency* perspective. The rationality lies across three dimensions. **1) For images**, spectral dependency represents the strong correlation between high-frequency image details and low-frequency structures, where higher-frequency components build upon lower-frequency foundations to progressively construct a complete image. **2) For models**, neural networks are verified to first fit the low-frequency information and then the harder high-frequency part Ulyanov et al. (2018). **3) Diffusion model essence.** Diffusion models are proved to be implicit approximate frequency autoregression in nature Dieleman (2024). With these insights, we propose the frequency progressive autoregressive (FAR) paradigm. Specifically, as shown in Figure 3, FAR applies spectral filters to get the corresponding images of different spectral bands. The autoregression is conducted along these spectral-filtered images form lower to higher frequency, thereby inherently satisfying the causality requirements of AR models. For each spectral-filtered image, FAR bidirectionally models the full token sequence, effectively preserving the spatial locality of image data.

To instantiate FAR with continuous tokenizer, we identify the challenges of optimization difficulty and variance when modeling image tokens at different frequency levels, and propose simplifying and re-weighting the diffusion modeling of these levels. Additionally, to enhance training and inference efficiency, we introduce the mask mechanism and frequency-aware diffusion sampling. These proposed techniques, coupled with the intrinsic harmony between spectral dependency and image data, endow FAR with a more compatible autoregressive paradigm for image generation. Comprehensive experiments on the ImageNet dataset demonstrate the efficiency and scalability of FAR, wherein it significantly reduces inference steps to only 10 while maintaining high quality and structural consistency. We also extend FAR to text-to-image generation. FAR achieves promising generation quality and high prompt alignment, utilizing much smaller model size, data scale, training compute, and inference steps, compared to previous text-to-image models.

Overall, our contributions can be summarized as follows:

- We propose the FAR paradigm, leveraging the spectral dependency of image data. FAR fits the causality requirement of AR models and preserves the spatial locality of image data, while being more sampling efficient.

- We delve into the instantiation of FAR with the continuous tokenizer, introducing a series of techniques to address the optimization challenges and improve the efficiency of both training and inference.

- We demonstrate the effectiveness and scalability of FAR through comprehensive experiments on ImageNet dataset and further extend FAR to text-to-image generation.

## 2 RELATED WORKS

### 2.1 TOKENIZERS IN AUTOREGRESSIVE MODELS

Most of existing AR models in the vision domain employ discrete tokens via vector quantization. The pioneering VQVAE Van Den Oord et al. (2017); Razavi et al. (2019) proposes to quantize the latent space with a finite codebook, where each original token is replaced with the nearest discrete token in the codebook. RQ-VAE Lee et al. (2022) proposes residual quantization (RQ). However, due to the significant information loss induced by quantization, the performance upper bound of AR methods may be limited. In contrast, continuous tokenizer Rombach et al. (2022) aligns more naturally with image data and induces less information loss. Some recent works Li et al. (2024); Fan et al. (2024); Deng et al. (2024) also propose to employ a continuous tokenizer for autoregressive generation. In this paper, we also leverage continuous tokenizer and integrate it with our FAR paradigm. Comprehensive analyses of the tokenizer are available in Section B of the Appendix.

### 2.2 AUTOREGRESSIVE MODELS FOR IMAGE GENERATION

Autoregressive model is an important method for image generation, leveraging GPT-style Radford (2018) to predict the next token in a sequence. Raster-scan flattens the 2-D discrete tokens into 1-D sequences in a row-by-row manner. Most of previous image autoregressive models employ this manner, including VQGAN Esser et al. (2021), VQVAE-2 Razavi et al. (2019), Parti Yu et al. (2022), DALL-E Ramesh et al. (2021), LlamaGen Sun et al. (2024), etc.

Besides this classical paradigm, some recent works make inspiring improvements over the raster-scan way. For example, MaskGIT Chang et al. (2022) proposes masked-generation to generate next token set instead of next one token, substantially reducing the inference steps. MAR Li et al. (2024) proposes to combine continuous tokenizers with mask-based generation. Another type of methods adopt residual quantization. For example, RQ-VAE Lee et al. (2022) proposes modeling the residual with vector quantization. VAR Tian et al. (2024) combines RQ-VAE with multi-scale, adding all scales to get the final prediction. This design enables the scale number to be the inference step. Direct compressing latent into 1-D sequence is also an interesting direction. TiTok Yu et al. (2024b) compresses images into 1-D sequences with a modified autoencoder design.

Some concurrent works Pang et al. (2024b); Yu et al. (2024a); Wang et al. (2024b); Pang et al. (2024a); Ren et al. (2024) propose other interesting AR methods. For example, RAR Yu et al. (2024a) combines randomness and raster-scan and progresses from randomness to raster-scan for sequence generation. FlowAR Ren et al. (2024) combines the multi-scale design and flow model, and proposes multi-scale flow model for image generation. Additionally, some multi-modal large models Zhou et al. (2024); Xie et al. (2024); Wang et al. (2024a) integrate the image generation ability into the AR models for unified understanding and generation.

Different from previous works, we propose FAR for autoregressive image generation with frequency progression, which fits the causality requirement of AR and preserves the unique prior of image data.

## 3 PRELIMINARIES

### 3.1 DIFFUSION LOSS FOR CONTINUOUS TOKENS

For the tokenizer in autoregressive models, the key is to model the per-token probability distribution, which can be measured by a loss function for training and a token sampler for inference. Following MAR Li et al. (2024), we adopt diffusion models to solve these two bottlenecks for integrating continuous tokenizer into autoregressive models.

**Loss function.** Given a continuous token $z$ produced by a autoregressive transformer model and its corresponding ground-truth token $x$, MAR employs diffusion model as loss function, with $z$ being the condition.

$$\mathcal{L}(z, x) = \mathbb{E}_{\varepsilon, t} \left[ \| \varepsilon - \varepsilon_\theta \left( x_t \mid t, z \right) \|^2 \right]. \tag{1}$$

Here, $\varepsilon \sim \mathcal{N}(\mathbf{0}, \mathbf{I})$, and $x_t = \alpha_t x_0 + \sigma_t \varepsilon$, with $\alpha_t$ being the noise schedule Ho et al. (2020). The noise estimator $\varepsilon_\theta$, parameterized by $\theta$, is a small MLP network.

Figure 2: Three prevailing regression direction paradigms in AR models for image generation. (a) Vanilla AR: sequential next-token generation in a raster-scan order; (b) Masked-AR: next-set prediction with random order, generating multiple tokens each step; (c) VAR: combines RQ-VAE and multi-scale, adding all scales to get the final prediction.

**Token sampler.** The sampling procedure totally follows the inference process of diffusion model. Starting from $x_T \sim \mathcal{N}(\mathbf{0}, \mathbf{I})$, the reverse diffusion model iteratively remove the noise and produces $x_0 \sim p(x|z)$, under the condition $z$.

### 3.2 REGRESSION DIRECTION

Regression direction plays a crucial role in autoregressive models for image generation. In Figure 2, we illustrate the three prevailing regression direction paradigms for image autoregressive models.

1) Vanilla AR (next-token prediction). The "next-token prediction" approach Esser et al. (2021); Sun et al. (2024) flattens the interdependent 2-D latent tokens via raster-scan. This paradigm, however, violates the causal requirements of AR sequences. For example, the tokens at the front of the next row should depend on the tokens near it, instead of the token at the end of the last row. Another limitation is the inference speed, demanding the token length as step, which is unbearably slow for high-resolution image generation.

2) Masked-AR (next-set prediction). The mask-based generation method Chang et al. (2022); Li et al. (2024) predicts the masked tokens given the unmasked ones. This paradigm enhances vanilla AR by incorporating randomness and predicting multiple tokens at every step. However, similar to the AR approach, Masked-AR violates the unidirectional dependency assumption of autoregressive models and neglects the image prior, limiting its potential.

3) VAR (next-scale prediction). VAR Tian et al. (2024) combines RQ-VAE Lee et al. (2022) with multi-scale, aggregating all scales to produce the final prediction. VAR maintains the spatial locality and adheres to the causality requirement. However, its multi-scale discrete residual-quantized tokenizer deviates from the commonly used tokenizers, necessitating specialized training. More importantly, we reveal that VAR paradigm demonstrates poor compatibility with the continuous tokenizer. Specifically, experiments combining VAR with the continuous tokenizer resulted in poor generation performance. Comprehensive results and analyses are provided in Section 5.

## 4 METHODOLOGY

### 4.1 SPECTRAL DEPENDENCY

The pivotal challenge for the regression direction of AR lies in harmonizing the causal sequence requirement with the inherent image prior. In this paper, we identify spectral dependency as a distinctive image prior tailored to this context. *The rationality of such design lies across three dimensions.* **1) For image**, it consist of low-frequency components that capture overall brightness, color, and shapes, alongside the high-frequency part that convey edges, details, and textures Russ (2006); Wornell (1996). The generation of higher-frequency information intrinsically relies on the prior establishment of the lower one. This hierarchical process also mirrors human artistic painting, where an initial sketch outlines the overall structure, followed by the progressive addition of details.

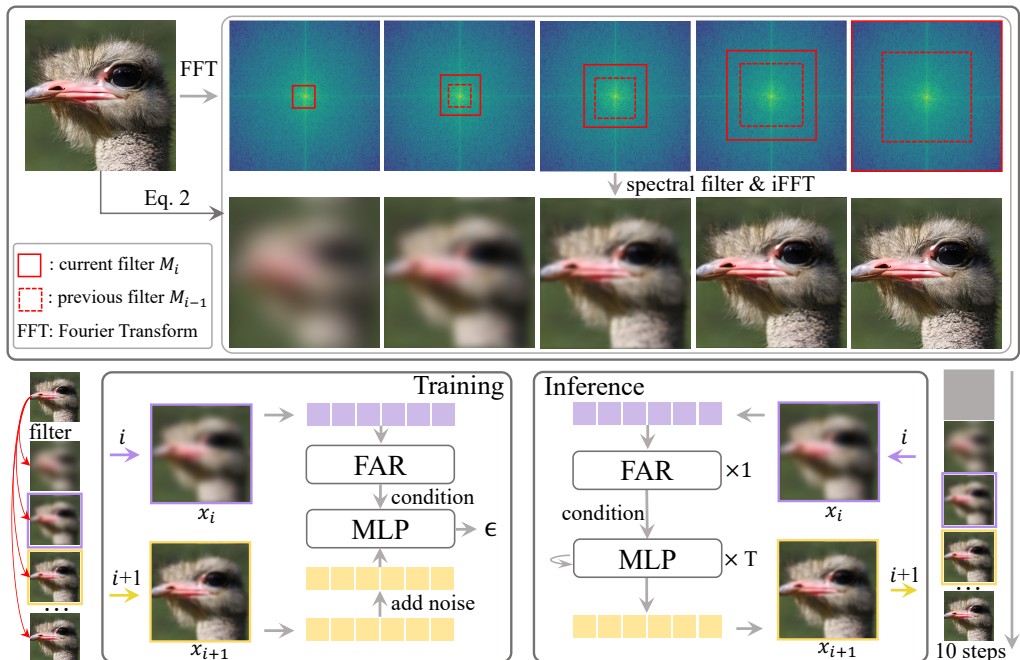

Figure 3: **Next-frequency prediction paradigm for autoregressive image generation via spectral dependency prior**. The upper part presents the spectral dependency and spectral filter process of images. The bottom part illustrates the training and inference stages of FAR.

**2) For models**, neural networks inherently exhibit similar spectral dependencies. DIP Ulyanov et al. (2018) found that neural networks demonstrate high impedance to high-frequency components while allowing low-frequency components to pass with low impedance. This indicates that neural networks prioritize learning low-frequency before progressing to the more complex high-frequency. **3) Diffusion model essence.** Diffusion model is known to model the distribution transition between noise and images via iterative noising and denoising. The recent work Dieleman (2024) further reveals that noising implicitly functions as spectral filter and diffusion is approximate spectral autoregression. This essence also inherently supports the optimality of our explicit FAR.

### 4.2 FAR: NEXT-FREQUENCY PREDICTION

Leveraging the spectral dependency, we introduce the innovative next-frequency prediction for autoregressive image generation. As shown in Figure 3, for each image $x$, its intermediate input at frequency level $i \in \{1, 2, .., F\}$ is formed as:

$$x_i = \mathcal{F}^{-1} M_i \mathcal{F} x. \tag{2}$$

Here, $F$ denotes the number of frequency levels. For the latent resolution of $16 \times 16$, we can obtain 16 spectral bands (F=16). $\mathcal{F}$ and $\mathcal{F}^{-1}$ represent the Fourier transform and the inverse Fourier transform, respectively. $M_i$ denotes the spectral filter within level $i$, and larger $i$ represents more clear image. $x_i$ is the corresponding spectral-filtered image.

During training, for each frequency level $i$ and corresponding input $x_i$, FAR employs bidirectional attention and predicts all tokens simultaneously at each step. This effectively models the dependencies between tokens in the 2-D plane, thereby preserving the spatial locality of images. The output of FAR works as the condition of the following MLP network, wherein $x_{i+1}$ functions as target.

During inference, FAR conducts autoregressive generation along spectral levels, progressively enhancing image clarity. The total steps for generating an image is reduced to only 10 (The inference step can be flexible as described in the following section). For each step, FAR forwards only once to get the condition of diffusion MLP, which then iterates T times (e.g. 100 in this paper) to get $x_{i+1}$.

### 4.3 FAR WITH CONTINUOUS TOKENS

In this section, we delve into the combination of FAR and continuous tokens. We first identify and solve two primary challenges: optimization difficulty and variance in modeling token distributions across different frequency levels. Then, we present two techniques to enhance training and inference efficiency. Visualization of the implementation details is available in Figure 7 of the Appendix.

**Optimization difficulty.** The diffusion loss in the continuous tokenizer models the distribution of per token. For FAR, diffusion loss needs to model $p\left(x_{i+1} \mid x_i\right)$ for $i \in [1, F-1]$, encompassing $F$ frequency levels. This multi-level distribution modeling is challenging for the relatively small MLP. To mitigate this, we propose to directly model $p\left(x \mid x_i\right)$, and then filter $x$ to get $x_{i+1}$. This simplifies the optimization complexity by normalizing the diffusion loss to only model $x$.

**Optimization variance.** Different frequency levels present varying optimization difficulties. Higher-frequency inputs are easier to predict, resulting in the optimization process being dominated by the more challenging low-frequency levels. To counteract this, we implement a frequency-aware training loss strategy that assigns higher loss weights to higher-frequency levels, ensuring balanced learning across all frequencies. Specifically, the loss weight is implemented in a sine curve, $w_i = 1 + \sin(\frac{\pi}{2} \times \frac{i}{F})$, where $w_i$ is the loss weight of frequency level $i$.

**Training efficiency.** During the early steps, FAR primarily needs to learn the low-frequency components, which are information-sparse. Consequently, utilizing all tokens is redundant. To this end, we propose to incorporate the mask mechanism into FAR, leveraging only a subset of tokens. Specifically, we devise a frequency-aware mask strategy that progressively increases the mask ratio for lower-frequency levels. The mask mechanism randomly masks $[r_i, 1]$ input tokens for the input tokens at frequency level $i$, where $r_i$ linearly transforms from 0.7 to 0. This design effectively reduces the training cost and we find it also contributes to improving generation diversity.

**Inference efficiency.** Diffusion model is known to first generate blurry structures first and then refines the details at later steps. The frequency progression property of FAR also inspires us to employ fewer diffusion sampling steps for lower frequency levels. Therefore, we devise the frequency-aware diffusion sampling step strategy that allocates progressively fewer steps to earlier frequency levels. Specifically, we linearly shift the sampling steps for $T = 40$ to $T = 100$, achieving an average sampling step of $T = 70$. This saves 30% inference time of the diffusion model.

## 5 EXPERIMENTS

### 5.1 SETUP

**Datasets.** For class-conditioned generation, we adopt ImageNet Deng et al. (2009) dataset. For text-to-image generation, we employ the JourneyDB Sun et al. (2023) dataset with ∼4.19M image-text pairs and ∼3.57M internal data. By default, all images are processed to 256×256 resolution.

**Training setup.** We use the AdamW optimizer ($\beta_1 = 0.9$, $\beta_2 = 0.95$) Loshchilov (2017) with a weight decay of 0.02. Unless otherwise specified, for class condition, we train for 400 epochs with a batch size of 1024, and at an exponential moving average (EMA) rate of 0.9999. For text condition, we train for 100 epochs with a batch size of 512 and EMA rate 0.99.

**Low-pass filters.** We explore two frequency filtering types: (a) first down-sample then up-sample in the spatial domain, (b) low-pass filter in the Fourier domain. We find that they yield similar performance, as shown in Section I of the Appendix. We empirically adopt type (a) for simplicity.

**Models.** We basically follow MAR Li et al. (2024) to construct our model, containing the AR transformer and diffusion MLP. The AR transformer has three model sizes: *FAR-B* (172M), *FAR-L* (406M), and *FAR-H* (791M). The diffusion MLP is much smaller as shown in Table 2. For text-to-image generation, we employ Qwen2-1.5B Yang et al. (2024) as our text encoder.

**Evaluation.** We evaluate FAR on ImageNet with four main metrics, Fréchet inception distance (FID), inception score (IS), precision and recall, by generating 50k images. For text-to-image generation, we adopt MS-COCO and GenEval Ghosh et al. (2024) dataset. FID is computed over 30K randomly selected image-text pairs from the MS-COCO 2014 training set. The GenEval benchmark measures the alignment with the given prompt.

Table 1: **Performance comparisons on class-conditional ImageNet 256×256 benchmark**. "↓" or "↑" indicate lower or higher values are better. FAR achieves comparable generation quality in nearly all the evaluated metrics compared to the sota methods, with only 10 inference steps. The only exception of FID is attributed to the slighter lower diversity, which we find to greatly influence the FID metric. The *Avg. Rank* in the last column represents the average ranking on the five indicators (including the additional inference steps) among the AR methods and our FAR-H. MAR (step=10) is evaluated using code and pretrained weights from their official GitHub repository.

| Tokenizer | Model | FID↓ | IS↑ | Precision↑ | Recall↑ | Params | Steps | Avg. Rank |
|---|---|---|---|---|---|---|---|---|
| **Diffusion Models** | | | | | | | | |
| Continuous | ADM Dhariwal & Nichol (2021) | 10.94 | 101.0 | 0.69 | 0.63 | 554M | 250 | - |
| | LDM-4 Rombach et al. (2022) | 3.60 | 247.7 | 0.87 | 0.48 | 400M | 250 | - |
| | U-ViT-H/2-G Bao et al. (2023) | 2.29 | 263.9 | 0.82 | 0.57 | 501M | 50 | - |
| | DiT-XL/2 Peebles & Xie (2023) | 2.27 | 278.2 | 0.83 | 0.57 | 675M | 250 | - |
| **Autoregressive Models** | | | | | | | | |
| Discrete | VQGAN Esser et al. (2021) | 15.78 | 74.3 | - | - | 1.4B | 256 | - |
| | ViT-VQGAN Yu et al. (2021) | 4.17 | 175.1 | - | - | 1.7B | 1024 | - |
| | RQTran. Lee et al. (2022) | 7.55 | 134.0 | - | - | 3.8B | 68 | - |
| | MaskGIT Chang et al. (2022) | 6.18 | 182.1 | 0.80 | 0.51 | 227M | 8 | 8 |
| | LlamaGen Sun et al. (2024) | 2.81 | 311.6 | 0.84 | 0.54 | 3.1B | 576 | 5 |
| | VAR Tian et al. (2024) | 3.30 | 274.4 | 0.84 | 0.51 | 310M | 10 | 2 |
| | PAR Wang et al. (2024b) | 2.88 | 262.5 | 0.82 | 0.56 | 3.1B | 51 | 6 |
| | RandAR Pang et al. (2024b) | 2.55 | 288.8 | 0.81 | 0.58 | 343M | 88 | 1 |
| | Open-MAGVIT2 Luo et al. (2024) | 3.08 | 258.3 | 0.85 | 0.51 | 343M | 256 | 4 |
| Continuous | MAR-H Li et al. (2024) | 1.55 | 303.7 | 0.81 | 0.62 | 943M | 256 | 2 |
| | MAR-H Li et al. (2024) | 9.32 | 207.4 | 0.71 | 0.47 | 943M | 10 | 9 |
| | **FAR-B** | 4.26 | 248.9 | 0.79 | 0.51 | 208M | 10 | 7 |
| | **FAR-L** | 3.45 | 282.2 | 0.80 | 0.54 | 427M | 10 | 3 |
| | **FAR-H** | 3.21 | 300.6 | 0.81 | 0.55 | 812M | 10 | 2 |

## 5.2 CLASS-CONDITIONAL IMAGE GENERATION

**Main results.** In Table 1, we list the comprehensive performance comparison with previous methods. We explore various model sizes and train for 400 epochs. Compared to most of the AR methods, our FAR is more efficient requiring fewer inference steps. Our method is superior to the VQGAN series with much smaller model size and inference steps. For recent works, like VAR and MAR, our method is also comparable in visual quality (indicated by IS and Perception metrics). Note that the lag in the FID metric is attributed to the slightly lower diversity (indicated by the Recall metric), which we find the FID metric is very sensitive to. Figure 1 shows qualitative results. We leave more visual results on ImageNet in Section G of the Appendix.

**Scaling of the autoregressive transformers and denoising MLP.** We investigate the scaling of both the autoregressive transformer and the diffusion loss model in Table 1 and Table 2. The autoregressive transformer takes the main burden of modeling the frequency dependency and mapping, thus also accounting for the majority of the parameters. We find that the size of FAR transformer significantly affects the performance. When scaling up the FAR transformer, the performance is consistently improved.

Table 2: **Scaling of denoising MLP in Diffusion Loss**. The denoising MLP is small and efficient, modeling the per-token distribution. Settings: FAR-L, 400 epochs, ImageNet 256x256.

| *MLP* | | | *Metrics* | |
|---|---|---|---|---|
| Depth | Width | #Params | FID↓ | IS↑ |
| 3 | 256 | 2M | 3.83 | 278.2 |
| 3 | 512 | 6M | 3.66 | 280.0 |
| 3 | 1024 | 21M | 3.45 | 282.2 |
| 3 | 1536 | 45M | 3.38 | 284.9 |

For the denoising MLP, the requirement to model only the per-token distribution, combined with our distribution modeling simplification strategy, allows a small MLP (e.g., 2M) to achieve competitive performance. As expected, increasing the MLP width helps improve generation quality.

**Sampling steps of FAR and diffusion loss.** The training of the FAR adopts the maximum of $F$ autoregressive step. For the inference, however, we can flexibly change the autoregression step and adopt fewer steps than $F$. Specifically, given $x_i$, FAR directly model $x$. In the next autoregressive step, we can filter $x$ to get a flexible next frequency level, i.e. $x_{i+2}$ instead of $x_{i+1}$, enabling the dynamic autoregression steps of FAR. Figure 4 depicts the generation performance under different FAR autoregression steps, where a higher step consistently achieves better performance.

Table 3: **Performance comparisons on text-to-image task.**. Metrics include MS-COCO zero-shot FID-30K and GenEval benchmark. Please note that FAR employs much smaller model size, training data, GPU costs, and inference steps. We do not intend to demonstrate that FAR achieves cutting-edge performance, but rather to verify its potential in achieving high efficiency and promising results.

| Tokenizer | Model | MS-COCO FID-30K↓ | GenEval | | | | | | | Params | Training Data | A100 Days | Infer Steps |
|---|---|---|---|---|---|---|---|---|---|---|---|---|---|
| | | | Sing-O. | Two-O. | Count. | Color | Pos. | Color-A. | Overall | | | | |
| **Diffusion Models** | | | | | | | | | | | | | |
| | LDM Rombach et al. (2022) | 12.64 | 0.92 | 0.29 | 0.23 | 0.70 | 0.02 | 0.05 | 0.37 | 1.4B | - | 6250 | 250 |
| Continuous | DALL-E 2 Ramesh et al. (2022) | 10.39 | 0.94 | 0.66 | 0.49 | 0.77 | 0.10 | 0.19 | 0.52 | 4.2B | 650M | - | 250 |
| | SD3 Esser et al. (2024) | - | 0.98 | 0.84 | 0.66 | 0.74 | 0.40 | 0.43 | 0.68 | 8B | - | - | 50 |
| **Autoregressive Models** | | | | | | | | | | | | | |
| | DALL-E Ramesh et al. (2021) | 27.50 | - | - | - | - | - | - | - | 12B | 250M | - | 256 |
| | CogView2 Ding et al. (2022) | 17.50 | - | - | - | - | - | - | - | 6B | 35M | - | - |
| Discrete | Muse Chang et al. (2023) | 7.88 | - | - | - | - | - | - | - | 3B | 460M | 2688 | - |
| | Parti Yu et al. (2022) | 7.23 | - | - | - | - | - | - | - | 20B | 4.8B | - | 256 |
| | LlamaGen Sun et al. (2024) | - | 0.71 | 0.34 | 0.21 | 0.58 | 0.07 | 0.04 | 0.32 | 775M | 60M | - | 256 |
| Continuous | **FAR** | 13.91 | 0.85 | 0.29 | 0.31 | 0.59 | 0.06 | 0.09 | 0.37 | 564M | 7.8M | 24 | 10 |

The training of the denoising MLP employs a 1000-step noise schedule following DDPM Ho et al. (2020). During inference, MAR verifies that fewer sampling steps ($T = 100$) are sufficient for generation. We further demonstrate that our frequency-aware diffusion sampling achieves comparable results with fewer steps. Specifically, we linearly shift the

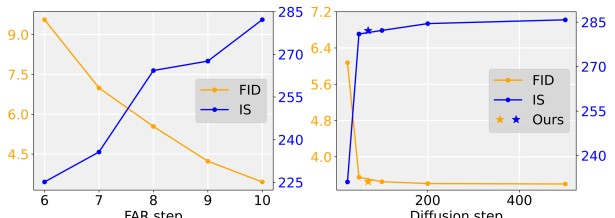

Figure 4: **Sampling steps of FAR and diffusion loss**.

sampling steps for $T = 40$ to $T = 100$, achieving an average sampling step of $T = 70$. This saves 30% inference time of the diffusion model. Figure 4 shows that our sampling strategy achieves comparable results with fewer steps.

**The compatibility of VAR and diffusion loss.** As we have noted in Section 3, the VAR paradigm demonstrates poor compatibility with the continuous tokenizer. The reasons are mainly two-fold. 1) The RQ manner is highly accuracy-sensitive for the prediction at every step for the continuous tokenizer. The RQ in VAR up-

Table 4: **Combining VAR and continuous tokenizer**.

| VAR components | | Metrics | |
|---|---|---|---|
| RQ | Multi-scale | FID↓ | IS↑ |
| ✔ | ✔ | 75.35 | 33.2 |
| ✗ | ✔ | 33.57 | 96.8 |

samples the prediction at each scale to the full latent scale and adds them all to get the final output. This requires highly accurate predictions at each scale. However, the exposure bias problem Bengio et al. (2015); Arora et al. (2022) of AR models induces inevitable error accumulation, deviating from the above requirement. 2) The per-token distribution modeling task in the VAR paradigm is prohibitively challenging for the diffusion loss. The tokens in different scales differ significantly in both the numeric range and receptive field. As shown in Table 4, directly combining VAR (RQ + mulit-scale) with the continuous tokenizer yields poor performance. Besides, we also try to remove the RQ design. The performance is improved due to no error accumulation in the residual paradigm. However, the performance still lags behind sota due to the gap in multi-scale. The visual results are available in Section D of the Appendix.

Note that FlowAR Ren et al. (2024) combines VAR with continuous tokenizer, via directly modeling the distribution of the whole image with multi-scale diffusion model (flow model), instead of the token-wise distribution. It is thus more like a multi-scale diffusion model than a AR model.

**More ablations.** We also conduct extensive ablations to verify the effectiveness of our method, including: **S1**) **DMS**: Diffusion loss Distribution Modeling Simplification strategy. **S2**) **Mask** mechanism. The mask mechanism improves the training efficiency by about 50%. **S3**) **FTL**: Frequency-aware Training Loss strategy. As shown in Table 6, with technique S1, FAR can already generate high-quality images (indicated by IS and Perception). The low FID is attributed to its lower diversity (indicated by Recall). With technique S2, the randomness in the mask mechanism compensates for

Table 5: **Ablations on the effectiveness of the proposed techniques**. Specific meanings of the abbreviations are in the ablation part. Settings: FAR-L, MLP size 21M, 400 epochs.

| Ablations | | | Metrics | | | |
|---|---|---|---|---|---|---|
| DMS | Mask | FTL | FID↓ | IS↑ | Pre↑ | Rec↑ |
| ✔ | ✗ | ✗ | 13.47 | 281.4 | 0.89 | 0.11 |
| ✔ | ✔ | ✗ | 4.11 | 288.9 | 0.79 | 0.51 |
| ✔ | ✔ | ✔ | 4.05 | 290.2 | 0.80 | 0.52 |

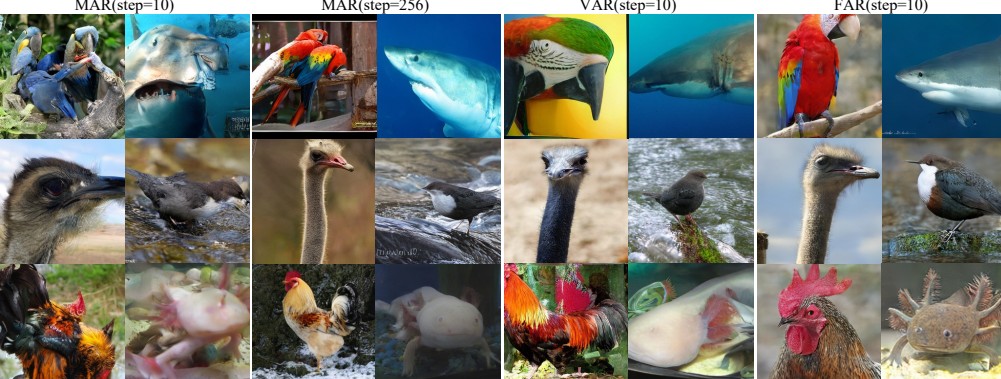

Figure 5: **Visual comparisons with the representative MAR and VAR methods with 10 inference steps**. Thanks to the intrinsic harmony with image data, our FAR can generate high-quality images with consistent structures and fine details with only 10 steps.

the diversity and thus achieves significant FID improvement. We depict visual comparisons of this ablation in Section F of the Appendix.

**Visual comparison with MAR and VAR.** As shown in Figure 5, the mask mechanism in MAR induces poor architecture under small inference steps. Adopting larger step elevates the generation quality but sacrifices the inference efficiency of this paradigm. The discrete tokenizer in VAR may also limit the performance limit and has difficulty in generating images with complex composition. In contrast, due to the intrinsic harmony with image data, FAR can generate high-quality images with consistent structures and fine details with only 10 steps.

### 5.3 TEXT-TO-IMAGE GENERATION

**Main results.** In Table 3, we depict the performance comparison on text-to-image generation task. Previous methods in this task usually employ substantially large model parameters, web-scale datasets, and unbearable computation costs. FAR can beat the classical DALL-E, CogView2 and LlamaGen, and achieve comparable performance to the recent sotas, with significantly smaller training and inference costs. The total training cost is 24 days with single A100. The concurrent works Fan et al. (2024); Deng et al. (2024) also verify the effectiveness of continuous tokens on text-to-image generation, but with substantially more training resources. In Figure 1 and Section E of the Appendix, we show the visual results on text-to-image generation. FAR can generate high-quality images with coherent structures and complex composition in 10 steps.

### 6 CONCLUSION

In this paper, we propose the frequency autoregressive generation paradigm and instantiate FAR with the continuous tokenizer. Specifically, we identify spectral dependency as the desirable regression direction for FAR. Besides, we delve into the integration of FAR and the continuous tokenizer. We demonstrate the efficacy and scalability of FAR through comprehensive experiments on class-conditional generation and further verify its potential on text-to-image generation.

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

# A    APPENDIX

This supplementary document is organized as follows:

Section B shows the comprehensive analyses on tokenizer from the perspective of compression.

Section C shows the details and visualization of the training and inference processes.

Section D demonstrates the poor compatibility of VAR and continuous tokenizer.

Section E shows more visual results of our method on text-to-image generation.

Section F depicts the visual results of ablations on the mask mechanism, which elevates generation diversity.

Section G presents the visual results as model size and inference step scaling up.

Section H shows the visual results at intermediate steps.

Section I shows the results of different low-pass filters.

Section J shows the prompts of the text-to-image generation for Figure 1 in main manuscript.

# B    COMPREHENSIVE ANALYSES ON TOKENIZER

**Tokenizer: discrete or continuous.** Data compression and reconstruction are vital for image generation, determining the performance upper bound of generation. Given the discrete nature and the mature categorical cross-entropy loss of languages, a commonly adopted strategy for visual autoregressive models is to discretize the data with VQ. However, compared to the discrete human-created language, natural image space is continuous and infinite. Quantization, specifically in VQVAE, inevitably introduces significant information loss.

For the tokenizer, the VQ operation induces significant information loss, making compression stage the bottleneck for better generation. Further, the autoregressive paradigm, i.e., "predicting next tokens based on previous ones", is independent of whether the values are discrete or continuous. The only difficulty that restricts the adoption of continuous-valued tokenizer is the lack of proper loss function to model the per-token probability distribution, which is easily done with cross-entropy for discrete-valued tokenizer. To this end, following the pioneering MAR Li et al. (2024), we adopt the diffusion model as loss function. Specifically, the autoregressive method predicts a vector for each token, which then serves as conditioning for the denoising network.

**Tokenizer: compression perspective.** For images, latent space is crucial for generation for the purpose of reducing computational burden. Thus besides the data format (discrete or continuous), we further analyses the tokenizers, VQGAN or VAE, as compression models from two key aspects: 1) Theoretical compression performance; and 2) Reconstruction visual results.

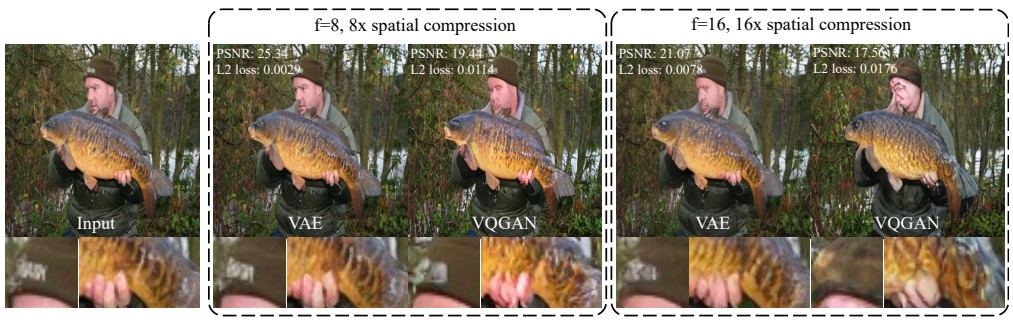

Figure 6: Image reconstruction performance comparison between continuous and discrete tokenizers under different spatial compression ratios (f=8 and f=16). Constrained by their finite vocabulary codebooks, discrete tokenizers suffer from significant information loss, struggling to faithfully reconstruct images with intricate, high-frequency details such as human faces. Note that the reconstruction of continuous tokenizer at f=16 is still better than the discrete one at f=8, which is also consistent with the rate distortion theory.

For measuring the theoretical compression performance, we adopt the information compression ratio (ICR) Chen et al. (2024). For discrete tokenizer, we take VQVAE as example, with downscaling factor $f$, codebook size $N$, input image's size of $H \times W$. We assume that the code follows a uniform distribution, so each code has $\log N$ bits information. For continuous tokenizer, we take VAE as example, with downscaling factor $f$, channel number $C$. Assume that the latent representation is fp32 tensor precision. The ICR of these two tokenizers are then as follows.

$$\text{ICR}(N, f) = \frac{(H/f) \times (W/f) \times \log N}{H \times W \times 3 \times \log 256} = \frac{\log N}{24 f^2}. \tag{3}$$

$$\text{ICR}(C, f) = \frac{(H/f) \times (W/f) \times C \times 32}{H \times W \times 3 \times \log 256} = \frac{32C}{24 f^2}. \tag{4}$$

Taking compression ration $f = 16$ for example, $\text{ICR}(N, f) = 0.23\%$ for discrete tokenizer with codebook size $N = 16384$ and $\text{ICR}(C, f) = 8.33\%$ for continuous tokenizer with channel number $C = 16$. Further, to achieve same ICR under same $f$, we need to exponentially enlarge the codebook size $N$ from 16384 to $2^{512}(1.34 \times 10^{154})$. Given that discrete tokenizers are inherently difficult to train Yu et al. (2023a); Mentzer et al. (2023), it is thus prohibitively hard to train codebook at this scale.

In Figure 6, we visualize the reconstructed images comparison between discrete and continuous tokenizers. Compared to continuous tokenizer, the discrete one has difficulty in both detail fidelity and semantic consistency. For instance, the character detail on the hat is poorly reconstructed by discrete tokenizer. For semantic, the face and fingers of the man as well as the fish scales fail to remain semantically identical by discrete tokenizer.

Based on the above analyses, the discrete tokenizer for images suffers from substantially more information loss than the continuous one, indicating that quantization, the shortcut stemming from mimicking languages autoregressive generation, may be a inferior solution for image data. Thus, apart from the commonly adopted VQ paradigm for image autoregressive models, it is quite necessary and promising to employ continuous tokenizer.

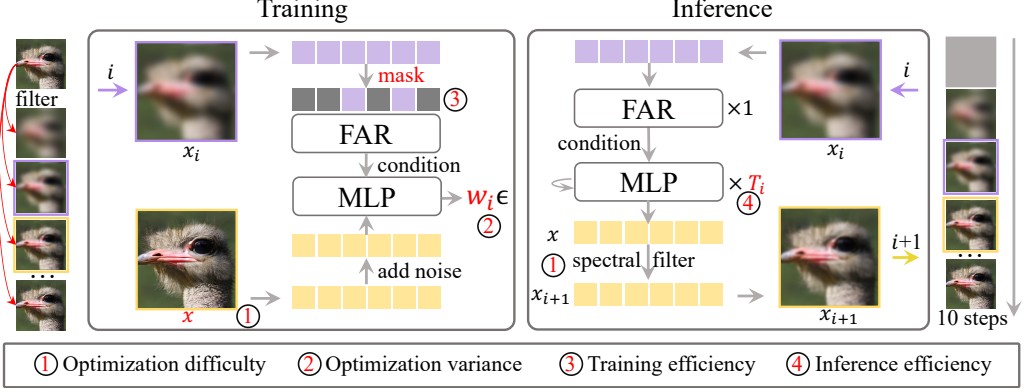

Figure 7: **The visualization of the training and inference processes of FAR.** This flow chart demonstrates the details of FAR and its integration with continuous tokens.

## C  DETAILS AND VISUALIZATION OF THE TRAINING AND INFERENCE PROCESSES

In Figure 7, we depict the flow chart of the training and inference processes of FAR, demonstrating the implementation details. *For the training process*, FAR randomly selects a frequency level $i$ and filters the input image $x$ into the intermediate frequency level $x_i$. FAR then adopts the mask ratio $[r_i, 1]$ to the token sequence. The output $z_i$ of the FAR model is then conditioned on the diffusion MLP. The diffusion loss models the distribution of each token with the frequency-aware dynamic loss weight $w_i$. *For the inference process*, we take the intermediate step $i$ as example. FAR takes the masked $x_i$ as input and outputs $z_i$. With $z_i$ as condition, the diffusion model samples the groundtruth token distribution $x$. Then, we can filter $x$ to get the next frequency level $x_{i+1}$.

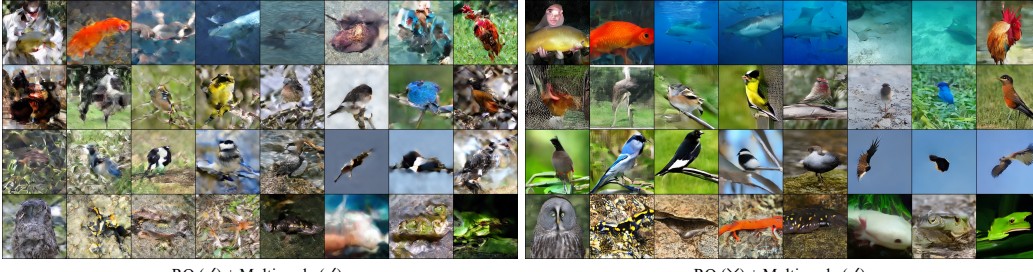

RQ (✓) + Multi-scale (✓)       RQ (✗) + Multi-scale (✓)

Figure 8: **The visual results of combining VAR and continuous tokenizer.** These images correspond to the first 32 class labels. Consistent with our analyses, VAR paradigm demonstrates poor compatibility with the continuous tokenizer. The generation suffers from poor architectures and severe artifacts. Removing the residual quantization helps reducing the artifact, but still suffers from poor image quality.

## D COMPATIBILITY OF VAR AND CONTINUOUS TOKENIZER: VISUAL RESULTS

In Figure 8, we depict the visual results of combining VAR and continuous tokenizer, corresponding to Table 4 of the main manuscript. The direct combination of VAR and continuous tokenizer demonstrates poor compatibility, generating images with obvious artifacts and poor architectures. On the right part, removing the Residual Quantization (RQ) successfully reduces the artifacts as expected. While, the generation is still inferior due to the challenging distribution modeling of diffusion loss in this case. These visual results exactly match our analyses in the manuscript.

## E MORE VISUAL RESULTS OF TEXT-TO-IMAGE GENERATION

In this section, we present more text-to-image generation results of our method in Figure 9.

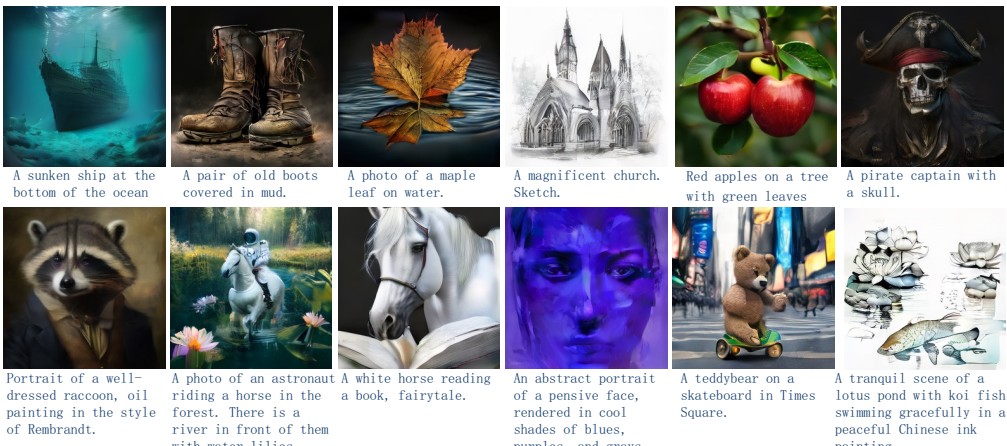

Figure 9: **Visual results of the text-to-image autoregressive generation at 256x256 resolution.**

## F VISUAL RESULTS OF ABLATIONS ON THE MASK MECHANISM

In Figure 10, we depict the visual results of the ablations on the mask mechanism, corresponding to Table 5 of the main manuscript. In the left part, our FAR can generate high-quality images after employing the diffusion loss distribution modeling simplification strategy. While, the generation diversity is limited. On the right part, we further adopt the mask mechanism. Mask introduces randomness, improving the generation diversity.

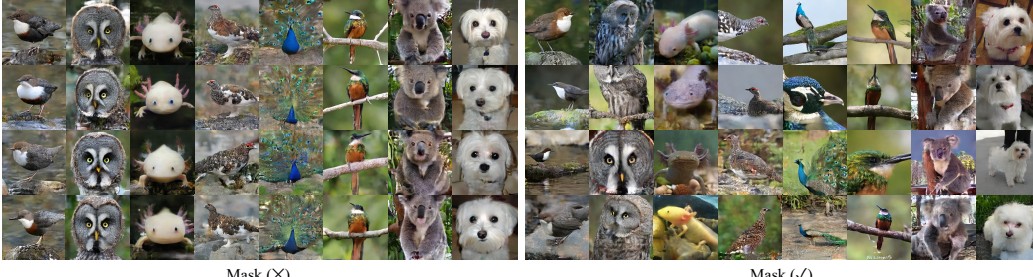

Mask (✗)    Mask (✓)

Figure 10: **The visual results of ablations on the mask mechanism.** Each column corresponds to one class label. Our FAR can generate high-quality images without mask mechanism, while suffers from low diversity within each class. The mask strategy can effectively improves the generation diversity.

## G    VISUAL RESULTS AS MODEL AND INFERENCE STEP SCALING

In Figure 11, we verify the scaling capacity of FAR: including model size and inference step. Scaling up these two factors can consistently improve the generation performance.

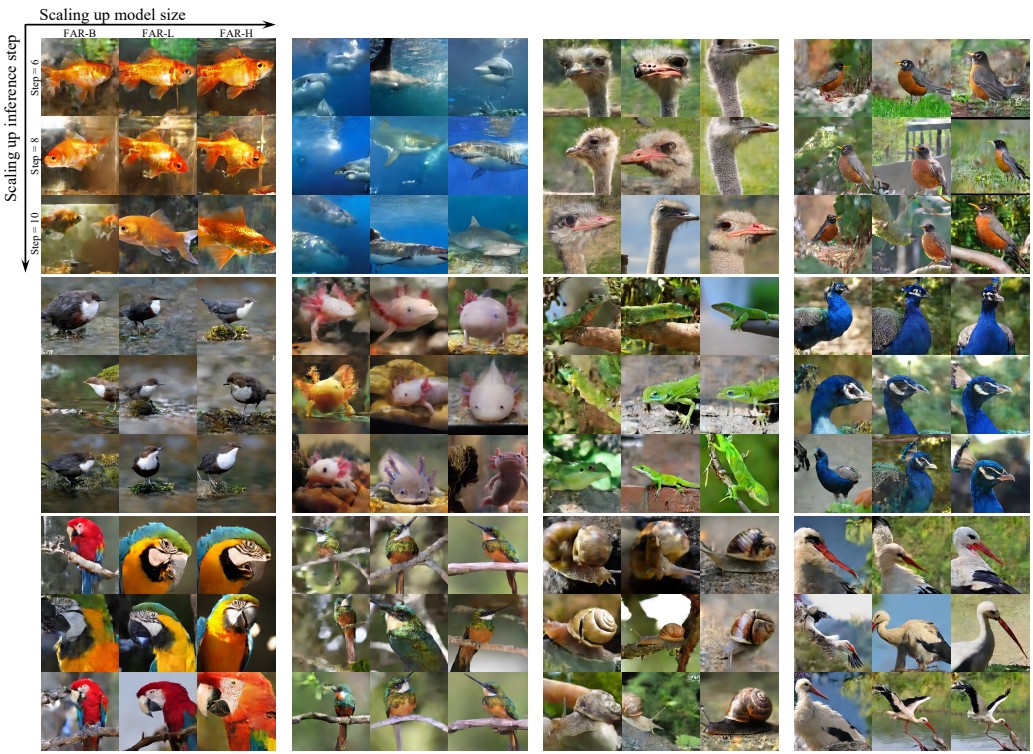

Figure 11: **Visual Results as Model and Inference Step Scaling.** We depict the generation results when increasing the model size and inference step. Scaling up these two factors can consistently improve the generation performance.

## H    VISUAL RESULTS AT INTERMEDIATE STEPS

In Figure 12, we present the intermediate generation results along the autoregressive generation process. In the early steps, FAR generates the overall color and structural information, and then refines the details.

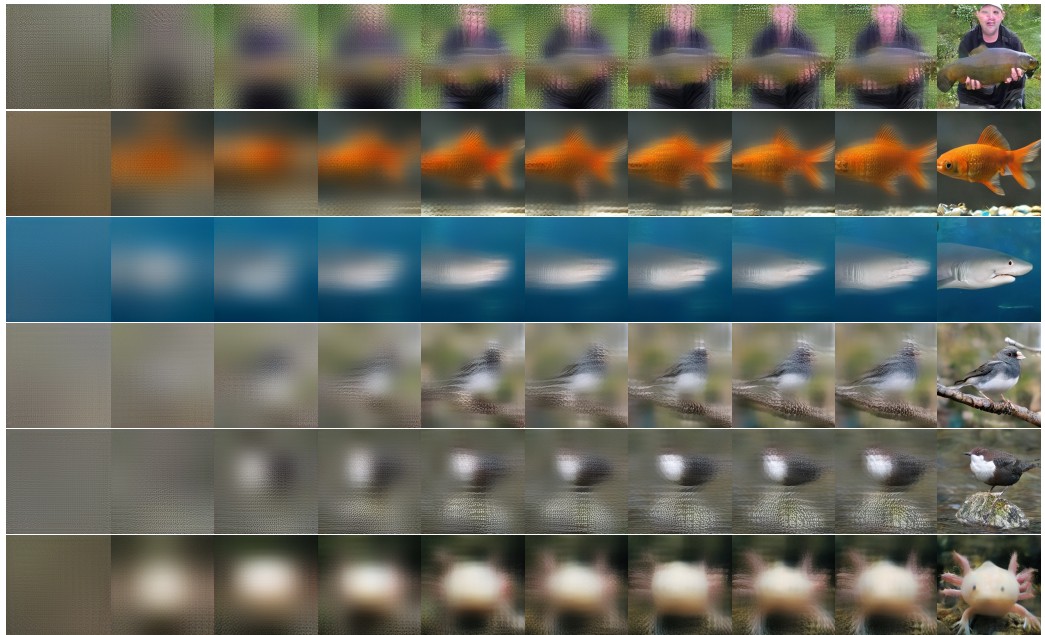

Figure 12: **Visual results at intermediate steps.** The intermediate generation results (total step 10) autoregressively refine the details, aligning perfectly with our frequency progression design.

## I DIFFERENT LOW-PASS FILTERS

We explore two frequency filtering types: (a) first down-sample then up-sample in the spatial domain, (b) low-pass filter in the Fourier domain. We find that they yield similar performance, as shown in Table 6. Since different frequency filtering methods only slightly differ in the filter. Besides, our method processes the filtered image in the spatial domain, which further narrows the difference of different low-pass filters. We thus hold that the frequency filtering methods make small differences to the final performance. By default, we empirically adopt type (a) for simplicity.

Table 6: Results under different low-pass filters

| Filters | FID↓ | IS↑ | Pre↑ | Rec↑ |
|---------|------|------|------|------|
| a | 4.05 | 290.2 | 0.80 | 0.52 |
| b | 4.21 | 291.3 | 0.80 | 0.51 |

## J PROMPTS FOR FIGURE 1 IN MAIN MANUSCRIPT

The following part presents the prompts for the text-to-image generation results in Figure 1 of the main manuscript:

- A mountain village built into the cliffs of a canyon, where bridges connect houses carved into rock, and waterfalls flow down into the valley below.
- An otherworldly forest of giant glowing mushrooms under a vibrant night sky filled with distant planets and stars, creating a dreamlike, cosmic landscape.
- A close-up photo of a bright red rose, petals scattered with some water droplets, crystal clear.
- A photo of a palm tree on water.
- A bird made of crystal.
- A tranquil scene of a Japanese garden with a koi pond, painted in delicate brushstrokes and a harmonious blend of warm and cool colors.
- Paper artwork, layered paper, colorful Chinese dragon surrounded by clouds.

- A still life of a vase overflowing with vibrant flowers, painted in bold colors and textured brushstrokes, reminiscent of van Gogh's iconic style.

- A peaceful village nestled at the foot of towering mountains in a tranquil East Asian watercolor scene.

- An enchanted garden where every plant glows softly, and creatures made of light and shadow flit between the trees, with a waterfall flowing in the background.

- A lion teacher wears a suit in the forest.

- A cloud dragon flying over mountains, its body swirling with the wind.

