# OpenReview forum: "Frequency Autoregressive Image Generation with Continuous Tokens"
_ICLR.cc/2026/Conference — ICLR 2026 Conference Withdrawn Submission_

### Official Review · Reviewer_Jp8c · 2025-10-25

**Soundness:** 2
**Presentation:** 3
**Contribution:** 2
**Rating:** 4
**Confidence:** 4

**Summary:**

This paper introduces Frequency Autoregressive Image Generation with Continuous Tokens (FAR), a new paradigm for autoregressive (AR) image generation. Unlike standard raster-scan AR methods that flatten 2D tokens into 1D sequences, FAR proposes a frequency-progressive generation process, in which the model autoregressively predicts higher-frequency image components conditioned on lower-frequency ones.

**Strengths:**

1. The idea of modeling image generation as a frequency-progressive autoregressive process is appealing.
2. FAR demonstrates fast inference (∼10 steps) and reasonable performance compared to large AR or diffusion models, which could be beneficial for practical deployment.
3. The paper is well-organized and clearly written. FAR is described in detail and the authors provide sufficient information about it.

**Weaknesses:**

1. Fourier components of an image are globally coupled and not conditionally independent. Thus, the assumed frequency progression seems does not constitute a valid autoregressive factorization.
2. The frequency masking and progressive reconstruction may cause high-frequency stages to modify or overwrite previously generated content, breaking causal consistency across stages.

**Questions:**

1. How is the autoregressive assumption justified mathematically in the frequency domain, given that Fourier components are globally dependent?
2. How does the model ensure consistency between low- and high-frequency layers during progressive generation?

---

### Official Review · Reviewer_5Mon · 2025-10-30

**Soundness:** 2
**Presentation:** 1
**Contribution:** 2
**Rating:** 2
**Confidence:** 3

**Summary:**

This paper introduces the frequency progressive autoregressive (FAR) paradigm, a variation of the image generation framework, along with a series of techniques to improve optimization and efficiency.
Motivated by the spectral dependencies present in images and neural models, this paradigm progressively generates images from low-frequency components to high-frequency components.
It employs a diffusion model to generate higher-frequency image components, conditioned on tokens produced from lower-frequency components. The diffusion model is implemented using a relatively small MLP, and the conditional tokens are generated from spectrally filtered images using bidirectional attention.
Given the nature of images, continuous tokens are adopted.
Performance comparisons were conducted on both unconditional and text-conditional tasks.
Ablation studies on several algorithmic design choices were also performed.

**Strengths:**

The topic addressed in this paper is important and interesting, as it explores new approaches to image generation. I agree that discussing the suboptimality of gradually generating images by patching and using autoregressive or masked-prediction methods is an important point.

**Weaknesses:**

At this moment, I am inclined to recommend rejection of this paper, as there are many unclear points.

1. The details of the proposed method are insufficiently described. The authors seem to rely on the algorithm of MAR [Li et al., 2024] in the Preliminaries section, which results in a lack of clarity. In particular, the explanation of the Token sampler is overly brief and should be elaborated.
2. Although the paper claims that there are only 10 steps, each step involves solving a diffusion problem with approximately 100 iterations, which appears inefficient. In practice, the process starts from T=40 at lower resolutions, averaging about 70 iterations per step, but this is still significant. Table 1 describes the number of steps as 10, but it would be more appropriate to use computational time as a metric. For clearer comparison, it would be helpful to show computational time in addition to the number of steps.
3. The explanation regarding low-pass filters around line 313 is inconsistent with Eq. (2). The description of down-sampling and up-sampling in the spatial domain does not involve the use of the Fourier transform, which contradicts the earlier explanation around line 259.
4. In the section "The compatibility of VAR and diffusion loss," the authors argue that VAR does not work well with continuous tokenizers, but they do not explain how continuous tokenizers are constructed for VAR or RQ. In fact, it is unclear how the continuous tokenizer is constructed from $x_i$, in Figure 3, in the main text.
5. According to the metrics in Table 3, FAR does not outperform other methods. Although the authors claim that FAR performs well despite using significantly less training data, it is difficult to fairly compare the methods because there are too many uncontrolled variables and blank entries in Table 3.
6. The paper does not appear to provide convincing evidence that continuous tokens are superior to discrete ones, even though this is considered to be one of the important claims of the work.


[Li et al., 2024] Li, Tianhong, et al. "Autoregressive image generation without vector quantization." Advances in Neural Information Processing Systems 37 (2024): 56424-56445.

**Other minor concerns:**
- The definitions of precision and recall in the Evaluation section are unclear. Presumably, they refer to https://arxiv.org/pdf/1806.00035, which should be cited. Providing references for FID and IS would also be preferable.
- The experiments use approximately 3.57M internal data, which raises concerns about reproducibility.
- The Avg. Rank in Table 1 is shown as an integer, which seems odd. Isn’t this metric the average of the rank in each column? Is it truncated?

**Questions:**

1. Could you provide a more detailed explanation of the methods, particularly regarding Weaknesses 1, 3, 4?
2. Regarding Weakness 2, could you present more direct evaluation metrics, such as computational time?
3. Regarding Weakness 5, is it possible to provide a more comprehensive and easy-to-understand comparison?
4. As mentioned in this paper, it is often stated that typical diffusion models gradually generate images from low-frequency components to high-frequency components [Yi et al., 2024]. In this context, what do you consider to be the critical differences between typical diffusion models and the proposed frequency progressive autoregressive (FAR) models? What are the principled benefits of the FAR paradigm?

[Yi et al., 2024] Yi, Mingyang, et al. "Towards understanding the working mechanism of text-to-image diffusion model." Advances in Neural Information Processing Systems 37 (2024): 55342-55369.

---

### Official Review · Reviewer_Pzvz · 2025-10-30

**Soundness:** 2
**Presentation:** 2
**Contribution:** 2
**Rating:** 2
**Confidence:** 5

**Summary:**

This paper introduces a novel image generation paradigm called Frequency Progressive Autoregressive (FAR), which replaces the traditional raster-scan "next-token prediction" method with spectral dependency. This new approach generates images by progressively building from low-frequency components to high-frequency ones, integrates this with a continuous tokenizer, and demonstrates its effectiveness on ImageNet and text-to-image generation tasks.

**Strengths:**

The model’s design ensures fewer inference steps, optimizing processing time and enhancing overall computational efficiency. This reduction directly translates to faster predictions, making it highly suitable for large-scale applications.

Well-Structured and Highly Effective Execution

**Weaknesses:**

While the proposed FAR paradigm aims to enhance sampling efficiency by reducing the number of generation steps, its practical performance remains notably limited when benchmarked against established autoregressive baselines. On ImageNet 256×256, for instance, the FAR-H variant (812M parameters, 10 sampling steps) attains an FID of 3.21 and an Inception Score (IS) of 300.6, whereas MAR-H [1](943M parameters, 10 steps) and SphereAR-H [2](943M parameters, 400 steps) achieve substantially better FID scores of 1.55 and 1.34, respectively, and higher IS values around 300–303. Even at comparable model sizes, SphereAR-B (208M) surpasses FAR-B (208M) with a FID of 1.92 vs. 4.26 and an IS of 277.8 vs. 248.9. These discrepancies, observed under equivalent data, resolution, and evaluation protocols, suggest that the efficiency obtained by reducing sampling steps fails to compensate for the deterioration in perceptual fidelity, diversity, and realism.

From a fairness standpoint, one would expect FAR to demonstrate competitive FID, IS, and precision/recall metrics under identical computational budgets (i.e., same number of FLOPs or wall-clock generation time) rather than solely relying on fewer sampling steps. Moreover, it would be informative to evaluate throughput–quality trade-offs (e.g., FID vs. sampling latency curves) or conduct human perceptual studies to substantiate the claim of efficiency without quality loss.

Beyond image quality, several conceptual aspects of FAR warrant further scrutiny. The reliance on spectral dependency modeling and continuous tokenization does not fully bridge the modality gap between natural images and language sequences, as it underrepresents spatial locality and structural priors inherent to visual data. Furthermore, while the frequency-domain representation is an interesting departure from raster-scan autoregression, the method currently lacks robust convergence analysis, ablation on frequency granularity, and scalability studies on higher-resolution datasets (e.g., ImageNet-512 or LAION subsets). Future work could explore integrating spectral modeling with hybrid spatial–frequency attention or adaptive frequency sampling to mitigate these weaknesses.

[1] Autoregressive Image Generation without Vector Quantization. NeurIPS 2024.
[2] Hyperspherical Latents Improve Continuous-Token Autoregressive Generation. Arxiv 2025.

**Questions:**

See weakness below

---

### Official Review · Reviewer_z3v3 · 2025-10-31

**Soundness:** 3
**Presentation:** 3
**Contribution:** 3
**Rating:** 4
**Confidence:** 5

**Summary:**

This paper proposes Frequency Autoregressive (FAR), a novel AR paradigm for image generation that leverages spectral dependency as the regression direction. Instead of raster-scan or random ordering, FAR progressively generates images from low-frequency to high-frequency components via Fourier-based filtering. The method integrates continuous tokens and employs diffusion-based loss for token distribution modeling. The authors introduce several techniques to address optimization challenges and improve efficiency, including distribution simplification, frequency-aware loss weighting, masking, and adaptive diffusion sampling.

**Strengths:**

1. Novel Paradigm: The core idea of frequency-progressive autoregression is elegant and well-justified from multiple perspectives: spectral dependency in natural images, the frequency bias of neural networks (DIP), and the implicit frequency autoregression nature of diffusion models.
2. Effective Integration of Continuous Tokens: The paper successfully combines continuous tokens with AR modeling, avoiding the information bottleneck of discrete VQ. The diffusion-based loss and proposed simplifications are practical and effective.

**Weaknesses:**

1. The confusing "Avg.Rank" in Table 1. There is a lack of a reasonable explanation for the basis on which the ranking is based. Are the weights of each of these indicators the same?
2. The generation capacity needs to be further enhanced. The performance of FAR in the FID metric is inferior to that of other autoregressive models, such as RandAR-XL(2.25) and RAR-L(1.7). Besides, these two models do not appear in Table 1.
3. Lack of exploration for higher resolutions. Both the c2i and t2i experiments in the paper were conducted at a resolution of 256*256. If there is only this one resolution setting, the superiority of FAR at higher resolutions cannot be demonstrated, which will affect its usability in other tasks.
4. Lack of inference time or a comparison of generation throughput. Although the paper repeatedly emphasizes that FAR only requires 10 steps of generation, it does not provide detailed inference time consumption or generation throughput, which raises doubts about the generation efficiency of FAR.

**Questions:**

1. Why is the maximum inference step set to 10? Is this a hard limit, or does performance saturate beyond this?

---

### Official Review · Reviewer_VoXQ · 2025-11-01

**Soundness:** 2
**Presentation:** 3
**Contribution:** 2
**Rating:** 2
**Confidence:** 4

**Summary:**

This paper introduces a new method for image auto-regressive (AR) modeling, termed FAR. FAR decomposes an image into components with different frequency bands and performs AR generation along the frequency dimension. Specifically, it employs FFT to obtain the image spectrum, applies spectral filters to isolate distinct frequency ranges, and then reconstructs the corresponding spatial images via inverse FFT for AR modeling. Furthermore, FAR incorporates a diffusion-based loss for AR modeling and introduces several techniques to improve training and inference efficiency. Extensive experiments on ImageNet-256 demonstrate the effectiveness of FAR compared with various existing AR paradigms. The authors also extend FAR to text-to-image generation.

**Strengths:**

1. The motivation of FAR is reasonable and intuitive — from the perspective of human perception, there indeed exists an inherent sequential relationship between the high and low frequency components of an image.
2. FAR introduces frequency-dependent schedules for both loss weighting and sampling steps.

**Weaknesses:**

My main concern lies in the novelty of the core idea and the insiginficant performance improvement.
1. Although the motivation is reasonable, it is not novel, as AR generation along the frequency dimension has already been explored by VAR [1]. Thus, the core idea of the paper is not original, and some claims regarding the novelty of the proposed method appear overstated.
2. The advantage of the propsoed framework is unclear. As the experimental results show, compared to VAR [1], the results of FAR are slightly worse while using a larger model and the same number of sampling steps.
3. Wall-time comparisons, rather than simply sampling steps, should be included in the main table. Since FAR employs a diffusion-based loss, which incurs additional computational cost compared to traditional classification-based sampling (e.g., VAR [1], LlamaGen [2]), more accurate efficiency comparisons with baseline methods are necessary for a fair evaluation.

[1] Tian et al., Visual autoregressive modeling: Scalable image generation via next-scale prediction.

[2] Sun et al., Autoregressive model beats diffusion: Llama for scalable image generation.

**Questions:**

1. As discussed in Appendix I, FAR adopts donw-sampling and up-sampling as the frequency filters, which is very similar to VAR. I am curious about the fundamental difference between FAR and VAR. Why can FAR be combined with a diffusion-based loss while Table 4 indicates that VAR fails to do so?
2. As shown in Table 1, the performance of MAR-H [1] drops significantly with only 10 steps. However, according to Figure 6 in the MAR paper [1], the FID does not decrease as sharply with 8 steps. Could the authors clarify the experimental settings used for MAR-H?

[1] Li et al., Autoregressive image generation without vector quantization.

---

### Note · Authors · 2025-11-14

I have read and agree with the venue's withdrawal policy on behalf of myself and my co-authors.